# Formation and Growth of Intermetallic Compounds in Lead-Free Solder Joints: A Review

**DOI:** 10.3390/ma15041451

**Published:** 2022-02-15

**Authors:** Mohd Izrul Izwan Ramli, Mohd Arif Anuar Mohd Salleh, Mohd Mustafa Al Bakri Abdullah, Nur Syahirah Mohamad Zaimi, Andrei Victor Sandu, Petrica Vizureanu, Adam Rylski, Siti Farahnabilah Muhd Amli

**Affiliations:** 1Faculty of Chemical Engineering Technology, Universiti Malaysia Perlis (UniMAP), Kangar 02600, Malaysia; izrulizwan@unimap.edu.my (M.I.I.R.); syahirahzaimi25@gmail.com (N.S.M.Z.); sitifarahnabilah@outlook.com (S.F.M.A.); 2Geopolymer & Green Technology, Center of Excellence (CEGeoGTech), Universiti Malaysia Perlis (UniMAP), Kangar 02600, Malaysia; 3Faculty of Materials Science and Engineering, Gheorghe Asachi Technical University of Iasi, D. Mangeron 41, 700050 Iasi, Romania; sav@tuiasi.ro; 4Romanian Inventors Forum, St. P. Movila 3, 700089 Iasi, Romania; 5National Institute for Research and Development in Environmental Protection INCDPM, Splaiul Independentei 294, 060031 Bucharest, Romania; 6Institute of Materials Science and Engineering, Faculty of Mechanical Engineering, Lodz University of Technology, Stefanowskiego 1/15, 90-924 Lodz, Poland; adam.rylski@p.lodz.pl

**Keywords:** intermetallic compound, alloying, surface finish, solder alloy

## Abstract

Recently, research into the factors that influence the formation and growth of intermetallic compounds (IMCs) layer in lead-free solders has piqued interest, as IMCs play an important role in solder joints. The reliability of solder joints is critical to the long-term performance of electronic products. One of the most important factors which are known to influence solder joint reliability is the intermetallic compound (IMC) layer formed between the solder and the substrate. Although the formation of an IMC layer signifies good bonding between the solder and substrate, its main disadvantage is due to its brittle nature. This paper reviews the formation and growth of IMCs in lead-free solder joints detailing the effect of alloying additions, surface finishes, aging time, aging temperature and solder volume. The formation and growth of the brittle IMCs were significantly affected by these factors and could be possibly controlled. This review may be used as a basis in understanding the major factors effecting the IMC formation and growth and relating it to the reliability of solder joints.

## 1. Introduction

Tin-lead (Sn-Pb) solder has been widely used as an interconnecting material for many years. Nevertheless, owing to health and environmental concerns, Pb usage in electrical equipment is prohibited [1,2]. The European Union’s (EU) directive on the Restriction of Hazardous Substances in Electrical and Electronic Equipment (RoHS) confirmed that the Pb usage in customer hardware is prohibited. Due to differences in their physical and mechanical characteristics, lead-free solder alloys cannot easily replace conventional lead-containing solder alloys. This is because, in comparison to Sn-Pb solder, lead-free solder systems usually create a thicker intermetallic compound (IMC) layer. Intermetallic phase generation in an electronic joint is normally undesirable due to the fact intermetallic phases are often brittle and more prone to crack growth.

IMCs are generated when two or more metallic elements come together to form a new phase having its own characteristics, crystal structure, and composition. The IMC phase could form in bulk solder during soldering application and at the interface between substrate and solder. Moreover, the solder alloy reacts with the substrate during the soldering process, forming an interfacial IMC layer at the interface. Its development has a substantial impact on the solder joint’s reliability [3]. In soldering, a thin coating of interfacial intermetallic is recommended to establish effective metallurgical bonding at the interface.

Nonetheless, owing to their brittle and hard character, excessive interfacial intermetallic compound development in solder joints is detrimental [4]. Therefore, the IMCs thickness layer, which defines the solder joint’s reliability and strength, is needed. A small amount of IMCs can be studied to signify that the solder joint has formed an excellent connection [5]. As a result, a persistent effort has been made to comprehend the mechanism of the interfacial interactions, which includes the intermetallic growth and formation at bulk as well as substrate/solder interfaces. Understanding the morphology, characteristics, and growth behaviour of intermetallic phases is critical for understanding solder connection reliability. Throughout the last several decades, researchers have examined the interaction between substrate and solder during soldering. For example, a recent study found that the expansion of the IMCs layer degrades the solder joint’s reliability.

This research examines the factors that govern the formation of lead-free solder IMCs with minor alloying elements and several types of surface finish to understand the performance of the interfacial IMCs layer. We also discuss the various studies on the effect of aging temperature as well as the volume of solder on IMC properties.

## 2. Effect of Alloying Additions

### 2.1. Formation and Growth of IMC in the Bulk Solder

The intermetallic phase might develop in the bulk solder as the main part of the eutectic phase and interfacial IMCs in a lead-free solder alloy. During the soldering process, some metal substrates dissolve in the molten solder, forming an interfacial IMCs layer at the metal-solder interface by a process known as dissolution within the solder and substrate. As the Cu level rises, the main Cu_6_Sn_5_ intermetallic species may be observed in the solder joint bulk. The Cu_6_Sn_5_ intermetallic is a key component of Pb-free solder joints’ microstructure. The microstructure of eutectic Sn-Cu alloy may be changed with the addition of a small amount of a given alloying element to the bulk microstructure of Sn-0.7Cu. Besides, the formation of intermetallics in solder alloy could be predicted by referring to the phase diagram, as depicted in Figure 1 [6].

Moreover, Nogita et al. [7] observed a considerable change in the nucleation pattern as well as behaviour of the Sn-Cu_6_Sn_5_ and Sn when up to 1000 ppm of Ni was added to Sn-0.7Cu solder. In the absence of Ni, Sn-0.7Cu shows a solidification process that progressed from the edge to the bulk solder alloy centre, a phenomenon is known as the ‘wall mechanism. In addition, the macroscopic development at the interface changed by adding Ni to this alloy, yielding a solid development from vast nucleation sites across the melting alloy. Gourlay et al. [8] also examined the impact of adding Ni to the Sn-Cu solder alloy, claiming that the transition in the solidification mode occurred at a concentration of 0–300 ppm Ni. The alloy’s microstructure was unaffected by raising the Ni concentration. Nogita et al. [9] introduced a small quantity of Ni to an Sn-Cu Pb-free solder and discovered that the IMCs layer in the Sn-Cu-Ni solder cracked less than the Ni state. In that work, synchrotron XRD was utilised to establish the Ni role in solder alloys, which is to stabilise the hexagonal (Cu, Ni)_6_Sn_5_. The nucleation and development of the main Cu_6_Sn_5_ intermetallic were considerably affected when 0.05 wt.% Ni was added to Sn-0.7Cu [10]. They also suggested that primary Cu_6_Sn_5_ developed at higher temperatures, which was smaller and more common in Sn-0.7Cu-0.05Ni/Cu joints compared to Sn-0.7Cu/Cu joints. Besides, Ventura et al. [11] investigated Sn-0.7Cu-Ni alloys with 0–1000 ppm Ni and discovered that Ni inhibits the production of β-Sn, lowering the volumetric proportion of the main β-Sn phase. Ni raises the interfacial energy among the eutectic phases, causing the eutectic to become irregular. Micro-additions of Ni can be used to control thtectic cellular prevalence development compared to a eutectic dendritic matrix in the Sn-0.7 wt.% Cu solder alloy, as described by Silva et al. [12]. As a result, small amounts of Ni (500–1000 ppm of Ni) appear to stabilise cell development, and if greater cooling rates are enforced, the creation of a dendritic array is achievable. The addition of Ni into Sn-Ag has been studied by Gumaan et al. [13]. They reported that the addition of Ni has been refined the particle size of β-Sn. Moreover, with 0.3 wt.% of Ni and distribution of Ag_3_Sn offer the potential benefit such as high strength, good plasticity and good mechanical performance.

According to Xian et al. [14], the addition of Al to Sn-0.7Cu might be employed to regulate the main size Cu_6_Sn_5_ rods, as illustrated in Figure 2. They hypothesized that the addition of a small amount Al into Sn-0.7Cu solder alloy helps to reduce the Cu_6_Sn_5_ size while increasing the number of Cu_6_Sn_5_ particles per unit volume. In Sn-0.7Cu/Cu and Sn-0.7Cu-0.05Al/Cu joints, the overall proportion of Cu_6_Sn_5_ is comparable. Apart from that, using synchrotron radiation X-ray imaging, Wang et al. [15] examined the development of Cu_6_Sn_5_ on Sn-6.5Cu. The growth behaviour of Cu_6_Sn_5_ at the Sn/Cu interface during soldering processes has been studied using synchrotron radiation. The authors found that by adding 0.2 Al to Sn-6.5Cu solder, the mean size of Cu_6_Sn_5_ was reduced compared to Sn-6.5Cu solder. This is due to the Al traces refinement on the Cu_6_Sn_5_ microstructures. The addition of Al interacted having Cu to create Cu-Al IMCs, inhibiting the development of Cu_6_Sn_5_. The Al influence on the Sn-0.7Cu solder alloys microstructure was investigated by Yang et al. [16]. Al refines the Sn-Cu solder alloy microstructure, according to the findings. Sn-Cu-Al solder alloy’s IMCs ranged from Cu_6_Sn_5_ in Sn-Cu-(0.01–0.025) Al to Al_2_Cu in Sn-0.7Cu-(0.05–0.075) Al. With increasing Al concentration, the volume fraction of the eutectic and IMC rises. According to Ma et al. [17], adding the rare-earth (RE) element La to the Cu-Sn solder alloy decreased the thickness of the Cu-Sn IMC interface layer of solder joints. This implies that the addition of modest quantities of La can minimize the driving power of Cu-Sn IMC formation.

Mcdonald et al. [18] also researched the effect of trace elements addition on the morphology and size of the primary Cu_6_Sn_5_ in Sn-4 wt.%Cu alloy with and without the addition of Ni. The investigation included additions of ppm of Pb, Ge, Ag, as well as Al. It was reported that Al influences the microstructure of the solder and refines the primary Cu_6_Sn_5_ size that occurs during primary solidification. They also claimed that the addition of Al increases the number of heterogeneous nuclei for Cu_6_Sn_5_. Reeve et al. [19] studied the Al growth in Sn-Ag-Cu and Sn-Cu. Every Al-modified alloy formed Cu-Al IMC particles having various phases and morphologies, according to the researchers. With rising Al concentration, a tendency of increased Cu-Al IMC volume percentage was discovered. This Cu-Al IMC nucleation between solder alloys benefitted the strength of the solder BGA systems. Apart from that, grain refinement in these joints and Cu-Al IMCs occurrence restrain the formation of a thermal fatigue crack. Small quantities of RE elements, primarily La and Ce, were added to the Sn-0.7Cu solder alloy by Wu et al. [20]. Adding 0.5%RE elements led to one-third to half of the β-Sn grains morphing into smaller 5–10 m. Precisely, adding 0.25%RE elements resulted in one-third to half of the β-Sn grains transforming into smaller grains of 5–10 µm. When 0.5%RE elements were added, the initial β-Sn grains were refined to the point where virtually all of them were 5–10 µm in size, and the solidified microstructure became evenly distributed Cu_6_Sn_5_.

Apart from that, the Zn and Ni addition effect on Sn-0.7Cu/Cu solder joints was studied by Zeng et al. [21]. The authors found that adding small quantities of Zn during solidification led to the production of a Cu-Zn intermetallic in the interdendritic region. However, adding small Ni amounts entirely alters the solidification mode, yielding a eutectic microstructure. Via Ni addition, tiny particles of primary (Cu, Ni)_6_Sn_5_ intermetallic formed ahead of the solidification front. The microstructure was modified by microalloying Zn and Ni, resulting in a more continuous, stable, and fine-grained interfacial Cu_6_Sn_5_ intermetallic while also reducing Cu_3_Sn development. Cu_6_Sn_5_’s polymorphic phase change is inhibited by Ni and Zn, which are homogeneously dispersed in the interfacial Cu_6_Sn_5_. In addition, Zeng et al. [22] improved the Sn-0.7Cu-0.15Zn solder’s solidification. During solidification, Zn has been shown to dramatically decrease the solidification range of β-Sn in an Sn-0.7Cu alloy. As a result, Zn might aid in the nucleation of β-Sn, which could impact phase selection during the near-eutectic alloys formation grown at low temperatures.

### 2.2. Formation and Growth of IMC at the Solder-Substrate Interface

For an interfacial IMC, the composition of the interfacial intermetallic is dictated by the combination of the metal surface and metal used. Interfacial intermetallic is necessary for every solder joint, and its reaction defines the overall solder joint’s reliability. During the reflow process between the substrate and the solder, a thin layer of interfacial intermetallic is generated. The Cu dissolution rate into the liquid, as well as the chemical reaction between Cu and Sn, are the key parameters that affect the thickness and form of the reaction layer. Tin-nickel (Sn-Ni) and tin-copper (Sn-Cu) are the two most frequent forms of interfacial intermetallic. The first phase of Cu-Sn interfacial intermetallic is formed nearest to the Cu interface and called “ɛ-phase” Cu_3_Sn intermetallic. Another phase of Cu-Sn, called “η-phase” Cu_6_Sn_5_, will form on top of the Cu_3_Sn intermetallic. The interfacial intermetallic formed between Ni and Sn at a much slower rate relative to the Cu-Sn intermetallic and is called the “δ-phase” Ni_3_Sn_4_ intermetallic. The Sn-Ni intermetallic is brittler than the Sn-Cu intermetallic [23,24].

The impact of alloying elements on the interfacial intermetallic has been studied by several researchers. These studies focus on the trace alloying elements added to the solder that can influence the formation and growth of IMCs [25,26]. Among alloying elements, Ni possesses the greatest impact on IMC growth. It reduces the thickness of Cu_3_Sn and the overall IMC layer [27,28]. Dissolution of Ni may affect the IMC layer on the Cu’s surface. The Cu atom in Cu_6_Sn_5_ can be substituted with Ni atom, which changes the composition to (Cu, Ni)_6_Sn_5_. Nagy et al. [29] elucidated the role of Ni in (Cu, Ni)_6_Sn_5_ IMC in Sn-Cu-Ni. They discovered that increasing the Ni content causes the Cu_6_Sn_5_ intermetallic’s lattice parameters a and c to decrease. The Ni addition to the lattice structure of Sn-0.7Cu causes a reflection shift at higher angles and a small change in the reflection intensity. Increased Ni concentration causes a reduction in mean crystallite size, according to these findings. The IMCs layer thickness for Sn-0.7Cu solder was larger compared to Sn-0.7wt.%Cu-0.3wt.%Ni solder, according to Rizvi et al. [30]. Besides, Nishikawa et al. [31] examined how an IMC formed and grew at the interface between a Cu pad and Sn-Ag(-Co) solders. (Cu, Co)_6_Sn_5_ and Cu_6_Sn_5_ are expected to be the intermetallic phases of the two regions. The replacement of Co atoms for Cu atoms in binary compounds with Sn is anticipated to generate a (Cu, Co)_6_Sn_5_ phase. They discovered that adding Co to the Sn-Ag solder had a significant impact on the IMC development and growth at the interface. Also, Sn-Ag-Co solders have an IMC thickness of about three times compared to binary Sn-Ag solders. The solders also had identical total IMC thicknesses after 1008 h of aging. The Sn-Ag-Co solder joint’s pull strength was insignificantly distinct compared to the binary Sn-Ag solder joint.

Moreover, according to Laurila, Vuorinen and Kivilahti [23], the rate of IMC growth and formation is influenced by the Ni concentration. The authors found that adding more wt.% Ni caused the intermetallic layer to expand faster. Therefore, the Cu_6_Sn_5_ compound was also created in the Sn-0.7Cu solder. Meanwhile, the (Cu, Ni)_6_Sn_5_ compound was formed in the Sn-0.7Cu-0.05Ni solder. The Ni addition to the Sn-Cu solder also reduced the production of cracks in the Cu_6_Sn_5_ IMC at the solder-substrate interface. Hence, as demonstrated in Figure 3, the stabilisation of the hexagonal Cu_6_Sn_5_ in Ni addition prevents volume variations that may trigger the cracking process [24]. The Sn-Ag-Cu with Au/Ni/Cu substrate was previously discovered by Yen et al. [32], who determined that the Ge and Ni addition did not affect the interfacial reactions. However, Ge segregation on the solder ball surface improved the anti-oxidation properties of the solder. Moreover, Wang and Shen [33] also studied the impact of Ni on the interfacial reactions between Ni substrate and Sn-Cu solders. Thus, the authors discovered that when the Ni concentration in the reaction phase grain of Cu_6_Sn_5_ rises, the morphology of the reaction phase grain of Cu_6_Sn_5_ shifts from a rod-like form to a faceted structure.

Nishikawa, Komatsu and Takemoto [31] examined how an IMC formed and grew at the interface between a Cu pad and Sn-Ag-Co solders. Here, (Cu, Co)_6_Sn_5_ and Cu_6_Sn_5_ are expected to be the intermetallic phases of the two regions. Moreover, the (Cu, Co)_6_Sn_5_ phase arises when Co is substituted for Cu in binary compounds containing Sn. Moreover, adding Co to the Sn-Ag solder had a substantial impact on the development and expansion of the IMC at the interface, according to the findings. As a result, the thickness of IMC layer in Sn-Ag-Co solder will be increased about three times compared to binary Sn-Ag solders. The solder link’s pull strength made with the Sn-Ag-Co solder was similar with respect to the binary Sn-Ag solder. Apart from that, Zeng, McDonald, Gu, Terada, Uesugi, Yasuda and Nogita [21] explained the effects of Zn and Ni in Sn-0.7Cu/Cu solder joints. Also, it demonstrates that slight Zn additions cause the creation of a Cu-Zn intermetallic in the interdendritic region during solidification.

Meanwhile, a slight quantity of Ni entirely alters the solidification mode and results in a eutectic microstructure. Small particles of primary (Cu, Ni)_6_Sn_5_ intermetallic form ahead of the solidification front when Ni is introduced. Microalloying Zn and Ni at the same time refine the microstructure, yielding a more continuous, fine-grained, as well as stable Cu_6_Sn_5_ intermetallic that inhibits Cu_3_Sn development. Cu_6_Sn_5_’s polymorphic phase change is inhibited by Zn and Ni, which are homogeneously dispersed in the interfacial Cu_6_Sn_5_. Apart from that, Zeng, Mcdonald, Gourlay, Uesugi, Terada, Yasuda and Nogita [22] investigated the solidification of Sn-0.7Cu-0.15Zn solder. It is widely known that during solidification, Zn may greatly limit the solidification range of β-Sn in an Sn-0.7Cu alloy. As a result, Zn may aid in the nucleation of β-Sn, which might impact phase selection during the development of undercooled near-eutectic alloys.

Furthermore, Zou and Zhang [34] investigated the impact of adding Zn to Sn-4Ag interacting with an Ag substrate. The authors discovered that increment in the Zn content in the Sn-4Ag solder causes the Ag-Sn IMC to become dominant at the interface, with the exception of the thinner Ag-Zn IMC. During the interfacial reaction with the increment of Zn concentration, Ag-Zn IMCs developed at the interface. Hence, the IMC shape changed from continuous IMC layers to discontinuous and loose IMC layers when the Zn concentration was increased. In addition, Chan et al. [35] created a novel solder alloy by mixing 0.3 wt.% Zn nanoparticles into Sn-3.8Ag-0.7Cu solder, resulting in thinner (Cu_3_Sn and Cu_6_Sn_5_) IMCs. When the extra percentage was increased to 0.8 wt.%, a new IMC layer (Cu_5_Zn_8_) was produced, increasing the overall IMC. On the other hand, Zhang et al. [36] found that adding 0.8% Zn to Sn-3.8Ag-0.7Cu solder slowed the formation of Sn-Cu IMC in the liquid/solid-state process owing to Zn atom accumulation at the interface.

In recent decades, Sn-Ag-Cu solder alloys became an option in substituting the lead solder for various applications due to their favourable thermal-mechanical fatigue as well as low melting temperature [37]. Moreover, adding alloying elements has a major effect on suppressing intermetallic phases’ growth. In summary, the addition of alloying elements on Sn-Ag-Cu has been summarized by Zeng et al. [38].

Shalaby [39] investigated the impact of adding Ni, Cr, and In to Sn-0.7Cu and discovered that the introduction of Ni, Cr, and In suppressed the development of eutectic quickly solidified Sn-0.7Cu alloy. Inside the Sn-rich phase, the synthesis of novel IMCs, which include In_3_Sn, Cu_6_Sn_5_, Cu_10_Sn_3_, and Ni-Sn, is more dispersed evenly. Li et al. [40] investigated the inclusion of indium to an Sn-0.7Cu-0.2Ni/Cu soldered joint. They reveal that eutectic Cu_6_Sn_5_ with a thickness of around 32.4 µm, and minor quantities of Sn are formed. The IMC layer thickness steadily grows as the amount of indium added rises. The IMC layer thickness grows from 36.2 to 55.6 µm when the indium inclusion is increased from 0.1 to 0.3 wt.%. Subsequently, Lee et al. [41] asserted that thermal aging of the Sn-Ag-Sb lead-free solder system causes the interfacial Cu_6_Sn_5_ compounds layer to shift to Cu_6_(Sn, In)_5_. Li et al. [42] investigated the introduction of phosphorus to Sn-0.7Cu. Also, the IMC thickness in the Sn-0.7Cu-0.005P solder joint appears to expand quicker compared to the Sn-0.7Cu solder joint. 

During creep-fatigue testing, the abundant IMCs may be the weakest part in the solder joint, and the crack may spread over the IMCs. They hypothesized that adding phosphorus to Sn-0.7Cu solder reduces its creep-fatigue life, probably due to increased gap production. The vacancies in the solder joint might contribute to crack starting or propagation areas. Also, Koo and Lee [39] reported the incorporation of Al to Sn-0.5Cu. Other than that, the Al inclusion promoted the creation of the Cu-Al (δ-Cu_33_Al_17_) IMC in the Sn-0.5Cu-based solder matrix that successfully prevented the production of eutectic β-Sn + Cu_6_Sn_5_ networks. The inclusion of Al disrupted the development of eutectic β-Sn + Cu-Sn IMC networks, and the quantity of Cu_6_Sn_5_ reduced as the quantity of Al was increased. Adding 0.02 and 0.2 wt.% Al to Sn-Cu reduced Cu_6_Sn_5_ nucleation undercooling and increased the quantity of Cu_6_Sn_5_ grains, according to Xian et al. [43]. Ma et al. [44] revealed that a Co-reinforced Sn-3.0Ag-0.5Cu composite solder inhibited IMC layer development. They hypothesized that the Co particles can bind Sn and Cu to produce Co-Sn or Co-Cu IMC near the interfacial layer, preventing Cu from being accessible for IMC layer creation and slowing their development rate. A minor Ga introduction, on the other hand, can stop IMC from growing. According to Luo et al. [45], Ga inclusion can lower the activity of Sn, hence suppressing the formation of Cu_6_Sn_5_. Gao et al. [46] explored how the introduction of Nd affects the evolution of the IMCs layer. The explanation for this might be due to the production of an Sn-Nd combination, which slows the Cu_6_Sn_5_ IMC growth. The Sn-Ag-Cu lead-free solder’s IMC thickness was diminished when a small trace quantity of RE elements La was introduced [47].

In their research, Zhang et al. [48] discovered that adding 0.03% Ce to Sn-Ag-Cu alloy decreased the IMC layer thickness and improved the solder joint strength. The morphology of IMC produced at the Sn-Ag-Cu/Cu and Sn-Ag-Cu-Ce/Cu interfaces eventually altered from scallop-like to planar-like, according to the researchers. In comparison to the Sn-Ag-Cu-Ce/Cu system, the Sn-Ag-Cu/Cu IMC grew at a faster pace. The Fe impact on the IMC growth kinetics during liquid state interfacial reaction has been examined, and Fe inclusion efficiently retards the interfacial Cu_6_Sn_5_ and Cu_3_Sn layer growth. The overall thickness of IMC Sn-Ag-Cu-Fe composite solder was comparable to Sn-Ag-Cu excluding Fe particles [49]. Haseeb and Leng [50,51] investigated the influence of Mo and Co nanoparticles on interfacial IMC between Cu substrate and Sn-3.8Ag-0.7Cu solder. They discovered that adding Mo nanoparticles to Cu_6_Sn_5_ interfacial IMC can reduce its thickness. The introduction of Co, on the other hand, can promote Cu_6_Sn_5_ development while suppressing Cu_3_Sn growth. As a result, they propose dissolving Co nanoparticles in Cu_6_Sn_5_ to alter the IMC compositions. As a result presented and analyzed above, it could be concluded that the adding of an alloying element will influence the IMC thickness and the IMC interfacial formation. Table 1 outlines the influence of minor alloying elements on IMC in the existing lead-free solder research database.

## 3. Effect of Surface Finish

The choice of a suitable surface finish is crucial because the IMC layer is considerably affected by the surface finishes material during soldering. The thickness and composition of IMCs also greatly affected by surface finish layers. To overcome the pace of growth between the solder and the substrate, several surface finishes are applied. Cu is the most widely utilized metal, given its excellent solderability. Even though Cu is the most frequent substrate for solder joints, various options work best. Electroless nickel immersion gold (ENIG), electroless nickel gold (NiAu), organic solderability preservative (OSP), and immersion Ag (ImAg) are the most prevalent surface finishes. The IMC layer is generally entirely formed on one of three metal surfaces: electroless nickel, electrolytic nickel or copper. The IMC layer formed on copper when soldering consists of Cu_6_Sn_5_ (close to the bulk solder). Also, Cu_3_Sn (close to the copper substrate) IMCs layer forms at the Cu_6_Sn_5_/Cu interface under high temperatures. Although the IMC layer generated on electrolytic nickel is Ni_3_Sn_4_, there is another intermetallic form present in this surface finish. The IMC layer generated on the copper surface finish can be affected by nickel dissolution in the solder [55,56]. The copper atom in Cu_6_Sn_5_ can be replaced with a nickel atom to form (Cu, Ni)_6_Sn_5_. (Cu, Ni)_6_Sn_5_ has been revealed to be more stable than Cu_6_Sn_5_ at room temperature where the content of Ni is ~9 at. % [57,58]. Before electroplating Sn, a Ni thin layer may be electroplated to provide a diffusion barrier between Cu and Sn, lowering the growth rate [55,59]. This is because the reaction between Ni and Sn is substantially slower at room temperature than the reaction between Cu and Sn. The response of IMCs with various surface finishes and solder alloys is discussed in this review. 

Yoon et al. [60] examined the influence of Cu/Ni content on the interfacial interactions between Sn-Ag-xCu solders and Cu/ENIG substrate. The creation and development of interfacial IMCs between a Ni-containing Sn-Ag-Ni solder and an ENIG substrate have been investigated. The outcomes contrasted with those of the Sn-Ag-Cu/ENIG system. The thickness of the Cu_3_Sn IMC was dramatically lowered when more Cu was introduced to the Sn-Ag solder, whereas the overall Cu-Sn thickness and Cu_6_Sn_5_ IMCs increased. At the Sn-3.0Ag-0.5Cu/ENIG and Sn-3.0Ag-0.5Ni/ENIG interfaces, (Ni, Cu)_3_Sn_4_ and Ni_3_Sn_4_ IMCs produced. The IMC layer for the Cu substrate grew 3.3 times quicker compared to the ENIG substrate in the Sn-3.0Ag-0.5Cu/Cu and Sn-3.0Ag-0.5Cu/ENIG joints, respectively. When soldered with Sn3.8Ag0.7Cu, Zheng et al. [61] discovered that immersion tin coating produced a smooth and homogeneous IMC layer when contrasted to copper plated with hot air solder levelling (HASL), OSP, or immersion silver. 

Moreover, Ourdjini et al. [62] investigated the IMCs formed during the soldering of lead-free Sn-Ag-Cu solder on copper (Cu), electroless nickel/ immersion gold (ENIG), immersion silver (ImAg), as well as electroless nickel/ electroless palladium/ immersion gold (ENEPIG) surfaces. When ENIG and ENEPIG soldering is completed, numerous morphologies of intermetallic having varied grain sizes occur at the solder joint interface, but when soldering on copper and immersion silver, a single intermetallic morphology form. The surface metallurgy utilized determines the type of intermetallic produced. It is generally known that when soldering on bare Cu and ImAg, Cu_6_Sn_5_ IMC is created during reflow. However, when soldering on ENIG and ENEPIG, (Cu, Ni)_6_Sn_5_ is produced. When comparing the grain sizes of the scallops, it is evident that thinner IMC is generated on Im-Ag than when soldering on Cu. In addition, Ha et al. [63] utilized Sn-Ag-Cu solder alloys to investigate the impact of various printed circuit boards (PCBs) between ENIG and ENEPIG. Hence, the IMC thickness in the ENEPIG board was less than that of the ENIG board. Furthermore, when the thermal aging period was increased, the IMC produced on the ENEPIG board developed slower than the ENIG board. 

The influence of Pd thickness on the interfacial response in solder joints between SAC and ENEPIG surface finish was investigated by Kim et al. [64]. In samples with 0.05 µm Pd thicknesses or lower, only (Cu, Ni)_6_Sn_5_ was found at the solder interfaces. Once the Pd thickness was raised to 0.1 µm or larger, the (Pd, Ni)Sn_4_ phase developed on (Cu, Ni)_6_Sn_5_. Moreover, Yoon et al. [65] examined the shear strength and interfacial reaction of eutectic Sn-0.7 wt.% Cu solder on Cu substrate and electroless Ni-P substrate. At the interface of the Sn-0.7Cu/Cu substrate, a Cu_6_Sn_5_ IMC was produced. The IMC produced at the interface of the electroless Ni-P substrate was (Cu, Ni)_6_Sn_5_. As a by-product of the Cu-Ni-Sn reaction, a P-rich Ni (Ni_3_P) layer was discovered between the (Cu, Ni)_6_Sn_5_ IMC layer and the electroless Ni-P layer. Also, the Ni_3_P, (Cu, Ni)_6_Sn_5,_ as well as Cu_6_Sn_5_, layer thicknesses were observed to rise with reflow time. Accordingly, the Cu_6_Sn_5_, (Cu, Ni)_6_Sn_5,_ and Ni_3_P layers were 1, 2.8, and 3.8 µm thick after reflowing at 255°C for 10 min. The (Cu, Ni)_6_Sn_5_ IMC layer grew quicker on the Cu substrate compared to the (Cu, Ni)_6_Sn_5_ IMC layer that grew on the electroless Ni-P substrate. With rising reaction time, the Ni content of the (Cu, Ni)_6_Sn_5_ IMC layer grew. The interfacial reaction of Sn-0.7Cu on Cu, Cu/Ni/Ag, Cu/Ni/Au, as well as Cu/Ni substrates are discussed by Yin et al. [66]. In the Sn-0.7Cu solder/Cu solder joint, the IMC layer was thicker compared to the Sn-0.7Cu solder Cu/Ni solder joint (Table 1). This is attributable to the fact that molten Sn-0.7Cu solder alloy on Ni substrate forms IMCs at a slower pace than it occurs on Cu substrate. Furthermore, given the quick chemical reactions of Ag-Sn and Au-Sn systems, the IMC layers average thickness at the Sn-0.7Cu solder/Cu/Ni interface is thicker compared to Sn-0.7Cu solder/Cu/Ni/Au as well as Sn-0.7Cu solder/Cu/Ni/Ag interfaces. This indicates that Sn-0.7Cu solder has higher wettability on Cu/Ni/Ag and Cu/Ni/Au substrates than Cu/Ni solder. 

Then, Aisha et al. [67] investigated the interfacial response between lead-free Sn-Ag-Cu-Ni solders and Cu, Im-Ag, and ENIG surface finishes during soldering. When soldering on ENIG surface finish, numerous morphologies of intermetallic with varied grain sizes occur at the solder joint interface. However, when soldering on copper and immersion silver, only one intermetallic morphology forms. Besides that, ENIG outperforms ImAg and Cu in regard to the impact of Ni addition on surface finish since ENIG’s IMCs are thinner than ImAg and Cu’s, implying that ENIG’s surface finish can decelerate Cu diffusion into Sn. Next, Yoon et al. [68] studied the interfacial response and joint reliability of Sn-3.0Ag-0.5Cu solder with ENIG and ENEPIG surface finishes. Compared to Sn-Ag-Cu/ENIG, the Sn-Ag-Cu/ENEPIG interface demonstrated greater strength and a smaller IMC thickness. In addition, the existence of the ENEPIG substrate’s thin Pd layer inhibited the development of the interfacial IMC layer and the Ni(P) layer consumption, culminating in improved solder joint interfacial stability.

Moreover, Sun et al. [69] looked at the interfacial response of Sn-0.4Co-0.7Cu, Sn-Ag-Cu, and Sn-Cu solder alloys on an immersion Au/electroless Ni(P)/Cu substrate. The analytical findings revealed that a tiny quantity of Co in the solder system would mostly react with Sn in the solder matrix to create CoSn_2_. In the Sn-Co-Cu solder, the Cu_6_Sn_5_ particles were the second IMC extensively dispersed. The finer Cu_6_Sn_5_ phase was more equally disseminated in the solder matrix than the rougher CoSn_2_ particles. For the Sn-Cu, Sn-Ag-Cu, and Sn-Co-Cu solders, the interfacial IMCs were 4.1, 4.3 and 3.5 µm thick, accordingly. In summary, the type of surface finish could affect the growth of IMC. Moreover, the higher reflow peak temperature and longer dwell time could increase the growth behaviour of IMC. Shen et al. [55] investigate the influence of isothermal aging and thermal cycling associated with Sn-1.0Ag-0.5Cu (SAC105) and Sn-3.0Ag-0.5Cu (SAC305) ball grid on ENIG and ImAg surface finish. They found that ENIG performs better than immersion Ag for applications involving long-term isothermal aging. SAC305 have a higher relative fraction of Ag_3_Sn IMC within the solder and performs better than SAC105 solder alloy. Table 2 displays the investigations on Sn-Ag-Cu solder alloy on the different surface finish.

## 4. Effect of Aging Time and Temperature

Recognizing the interfacial intermetallic reaction kinetics between the solder and the substrate in the electronic assembly context will help us extend the service life of the solder joints. Studying the impacts of aging time and temperature on the characteristics of interfacial intermetallics is thus critical. Cu_6_Sn_5_ and Cu_3_Sn phases are frequent interfacial layers between the Cu substrate and the solder during the soldering of an Sn-Cu solder system. Interfacial IMC layers are formed by two primary reactions: (i) dissolving the metal substrate in molten metal as well as (ii) connecting the active component elements in the solder material with the substrate material [80]. The formation of Cu_6_Sn_5_ occurs the reaction of molten solder and the substrate [81]:5 Sn + 6 Cu → Cu_6_Sn_5_
(1)

Meanwhile, Cu_3_Sn IMC can be formed either through the decomposition of Cu_6_Sn_5_ as in reaction 2 and 3 or through the consumption of Cu (reaction 4) during solid-state diffusion process [81]:Cu_6_Sn_5_ + 9 Cu → 5 Cu_3_Sn(2)
Cu_6_Sn_5_ + 3 Sn → 2 Cu_3_Sn(3)
3Cu + Sn → Cu_3_Sn(4)

Understanding the solder-substrate interface bonding response is significant to decide the reliability of solder interconnections. Investigators researched the development behavior of interfacial IMCs during solid-state annealing to improve solder coating reliability. The intermetallic phase is controlled directly by the solid-state diffusion process. The IMC layer will develop in the solid-state, causing solder joint reliability to deteriorate with time. The IMC layers grow gradually, which decreases defects and ductility such as voids at the interface, which could translate into failure for the solder interconnects. Under the aforementioned thermal conditions, the present IMC in solid joints will remain to expand through solid-state diffusion. An empirical power-law may be utilized to explain the isothermal growth kinetics of a continuous IMCs layer overall:d − d_o_ = Dt^n^
(5)
in which n denotes the time exponent, t denotes the reaction time, D denotes the growth rate coefficient, d denotes the aged thickness after time t, and d_o_ denote the as-soldered thickness of the reaction layer. It has been proven that a diffusion mechanism is primarily accountable for the IMC layer formation of many solder joints with time exponents of ~0.5 [82]: (6)$$D=\mathrm{Do}\;\exp(\frac{-Q}{\mathrm{RT}}),$$
in which T denotes the aging temperature, R denotes the gas constant, Q denotes the activation energy, and D_o_ denotes the growth rate constant in the absolute unit. Moreover, the activation energy may be estimated from the plot’s slope between 1/T and ln K employing the linear regression model. The IMC layer produced utilizing Sn-based solder on the Cu substrate represents typical Cu_6_Sn_5_. Various IMC can occur in addition to Cu_6_Sn_5_, relying on the solder compositions as well as surface finishes on the substrate. Cu_3_Sn intermetallic can be generated during high-temperature aging when the solder is deposited on the Cu [83], in addition to the Cu_6_Sn_5_. Cu_3_Sn usually observed at the ageing temperature above 150 °C [81]. The activation energy of Cu_6_Sn_5_ is lower than that of Cu_3_Sn, which implies that Cu_6_Sn_5_ will develop quicker than Cu_3_Sn, as per the Arrhenius relationship. As a result, inhibiting the development of Cu_6_Sn_5_ will be more successful than limiting the development of Cu_3_Sn [80]. Following the soldering step, the IMCs layer remains to expand, and the greater the temperature, the quicker the IMC layer grows. The IMC generated in Sn-3Ag-0.5Cu, and Sn-3Ag-0.5-Cu-0.06Ni-0.01Ge solder BGA packages having Ag/Cu pad is discussed by Chuang et al. [84]. They discovered that after aging at 150 °C, an extra Cu_3_Sn IMCs layer forms at the interface between the Cu_6_Sn_5_ and Cu pad in the Sn-3Ag-0.5Cu specimen. In the Sn-3Ag-0.5Cu-0.06Ni-0.01Ge specimen, though, the similar sort of intermetallic development is blocked. The isothermal aging of the Sn-0.7Cu solder/Ni BGA joint has been studied by Yoon et al. [85]. The interaction between the Ni layer and molten solder established a (Cu, Ni)_6_Sn_5_ layer at the interface during soldering. Only (Cu, Ni)_6_Sn_5_ was found since the sample was aged at 70–150 °C. The solder/Ni interface revealed a duplex structure of (Cu, Ni)_6_Sn_5_ as well as (Ni, Cu)_3_Sn_4_ after 50 days of isothermal aging at 170 °C, as depicted in Figure 4. Cu diffusion into the interface is reduced, resulting in the creation of the (Ni, Cu)_3_Sn_4_ IMC. 

Bi, Ju and Lin [3] examined the impact of introducing elemental Bi and Er to a low Ag solder alloy Sn-x(1.0, 1.5, 2.0)Ag-0.3Cu-3.0Bi-0.05Er on the development of the interfacial IMC layer (wt.%). The IMC layer thickness at the interface of a Cu/SACBE/Cu joint was discovered to be much less than that of a Cu/SAC/Cu joint. Also, the inclusion of Bi and Er successfully prevented the interfacial IMC layer from overgrowing during thermal aging. Furthermore, Hodulova et al. [86] studied the influence of Bi and In inclusions on the development of intermetallic phases in lead-free Sn-3.7Ag-0.7Cu solder joints. The joints were then matured in a convection oven at temperatures varying from 130 to 170 °C for 2 to 16 days. During soldering, Cu_6_Sn_5_ is created, while Cu_3_Sn is formed during solid state aging. Besides, in the lead-free solder, the growth rate of Cu_3_Sn was slowed by Bi and In inclusions. Cu_3_(Sn, In) or Cu_3_(Sn, Bi) compounds can occur at Cu_3_Sn grain boundaries, in which they restrict Sn diffusion, seeing as Bi and In can replace Sn in IMCs. At greater concentrations, indium is thought to promote copper diffusion in Cu_6_(Sn, In)_5_ and contribute to the creation of In-based compounds. In addition, Lejuste et al. [87] investigated the introduction of In to Sn-3.0Ag-0.4Cu-7.0In composite solder with Cu substrate. Two IMC layers, Cu_6_(Sn, In)_5_ as well as Cu_3_(Sn, In), were produced at the interface, according to the findings. They discovered that Cu_3_(Sn, In) has a reduced growth coefficient than Cu_6_(Sn, In)_5_, implying that the inclusion of the In element can delay the formation of the IMC layer during thermal aging.

Additionally, Wu et al. [88] examined the IMC growth kinetics impact between Sn-3Ag-3Bi-10In and Cu substrate at temperatures of 120, 150, and 180 °C, as well as aging times of up until 40 days. This results in Cu_6_(Sn,In)_5_ and Cu_3_(Sn,In) appearing as IMC layers at the interface, and they discovered that the IMC thickness of reactive layers did not vary greatly with aging time at 120 °C and 150 °C. However, it abided by a diffusion-controlled mechanism at 180 °C. Also, Kanlayasiri and Sukpimai [89] studied the influence of indium on the development of the interfacial layer on SAC0307. Indium had no influence on the IMCs layer thickness, according to the findings.

Furthermore, the inclusion of indium in the solder inhibited the formation of the Cu_3_Sn layer by lowering the Cu_6_Sn_5_ to Cu_3_Sn conversion. The activation energy of Cu_3_Sn rose as the indium level rose. Moreover, Wang et al. [90] also studied the introduction of Zn to the Sn-0.7Cu solder alloy. Small quantities of Zn added to the solder have been shown to inhibit the development of the IMC interface during thermal aging. These findings indicate that adding Zn to the Sn-0.7Cu alloy enhances the mechanical characteristics and reliability of the solder joints. In addition, Zeng, Xue, Zhang, Gao, Dai and Luo [38] explored the insertion of Zn atoms into IMCs as a means of inhibiting IMC particle development during thermal aging. Thus, (Cu, Ni)_6_Sn_5_ IMCs became (Cu, Ni, Zn)_6_Sn_5_ IMCs after Zn was added to Sn-0.7Cu solder. During thermal aging at 150 °C, they discovered that the (Cu, Ni, Zn)_6_Sn_5_ IMCs are more stable than the (Cu, Ni)_6_Sn_5_ and Cu_6_Sn_5_ IMCs. 

Alternatively, Wang et al. [91] studied the impact of moderate Ni, Co, and Fe additions on the reaction between the Sn-Ag-Cu solder and the Cu substrate. Also, the researchers discovered that both Cu_6_Sn_5_ and Cu_3_Sn phases developed during solid-state aging. The inclusion of Ni, Co, and Fe, on the other hand, resulted in a substantially thinner Cu_3_Sn layer. In addition, Rizvi, Chan, Bailey, Lu and Islam [52] investigated IMC production of Sn-2.8Ag-0.5Cu during isothermal aging with the inclusion of 1 wt.% Bi. It was discovered that adding 1 wt.% Bi to the Sn-2.8Ag-0.5Cu solder prevents the development of unwanted IMCs. Hence, the IMC between the Sn-2.8Ag-0.5Cu solder and the Cu substrate similarly changes morphology from scallop to planar.

On the other hand, Dong et al. [92] investigated the impacts of tiny amounts of RE elements on Sn-58Bi and Sn-58Bi-Ag. In conclusion, the IMC thicknesses of the solder joint were raised during high-temperature aging. Table 3 outlines the link between the thickness of the IMC layer and the aging time of the solder alloy. 

## 5. Effect of Solder Volume

The effect of solder volume has been investigated by several researchers to comprehend the IMC thickness at the solder joints [101,102,103,104]. Different solder volumes give rise to different interfacial reactions which in turns affect the formation of interfacial IMC and the microstructure [101]. Microstructure anisotropy and Joule heating could triggered the failure in microbumps [101]. The volume effect between Ni and Sn-Ag-Cu has been investigated by Yang et al. [105]. Solder spheres of three different sizes (300, 500, and 760 µm in diameter) were employed, with three distinct Cu concentrations (0.3, 0.5, and 0.7 wt.%). They investigated how Cu concentration in solder, solder volume, as well as temperature influenced soldering reactions between Sn-Ag-Cu solder and Ni during bump metallization. The kind of IMC evolves from (Cu, Ni)_6_Sn_5_ to (Ni, Cu)_3_Sn_4_ when the remaining visible Cu concentration in the solder reduces and the solder volume decreases. They also verify that when the temperature drops, the crucial Cu concentration diminishes. The Cu concentration gets increasingly complicated to maintain as the solder volume decreases. Besides, Chang et al. [106] explored how solder volume and Cu concentration affected Cu pad consumption during reflow soldering. Only Cu_6_Sn_5_, with the typical scallop-type morphology, were developed at the interface for all experimental circumstances, including Sn-3.0Ag and Sn-3.0Ag-0.7Cu, 400–960 µm solder balls, with 1–5 reflow cycles. By raising the original Cu concentration, expanding the reflow number cycles, and lowering the solder volume, the scallop sizes grew. As demonstrated in Figure 5, a reduction in solder volume facilitated the production of bigger Cu_6_Sn_5_ scallops. This was due to the fact that the Cu concentration in the solder grew quicker in smaller solder balls than in bigger ones. According to the Cu concentration impact, the scallops kept growing as the smaller balls were exposed to greater Cu concentrations for extended amounts of time. In their investigation on SAC405 solders reacting with EN(B)EPIG surface finish, Azlina et al. [107] obtained comparable findings. They reported that after reflow soldering, the average thickness of the intermetallic for bigger solders is thicker than for smaller solders. In the meantime, since the low concentration of Cu atoms is accessible in a small solder ball, smaller solder sizes formed thicker intermetallic than bigger solders after reflowing process.

Furthermore, Liu et al. [108] studied the effect on the interfacial reaction between Cu pads and Sn-3.0Ag-0.5Cu solder balls. Solder balls with diameters of 200, 300, 400, and 500 µm were selected. The researchers discovered that the lower the solder ball volume, the greater the interfacial Cu_6_Sn_5_ grains and the Cu_6_Sn_5_ IMC layer thickness. The dissolving of the Cu pad into molten solder was linked to the formation of the Cu_6_Sn_5_ IMC layer. Cu_6_Sn_5_ grains were finer interfacial due to the faster Cu dissolving rate in molten solder. Additionally, Ourdjini et al. [109] examined the Sn-Ag-Cu on electroless nickel/immersion gold (ENIG) surface finish with varied solder volumes (300, 500, and 700 µm). The findings demonstrated that the volume of solder utilized has a substantial impact on the intermetallic thickness produced at solder joints. Also, the intermetallic generated in smaller-volume solders was found to be thicker compared to those produced in larger-volume solders. The solder bump volume, on the other hand, has no bearing on the intermetallic kinds or morphologies. Specifically, (Cu, Ni)_6_Sn_5_ and (Ni, Cu)_3_Sn_4_ were found in Sn-Ag-Cu solder joints.

On the other hand, Huang et al. [110] employed an Sn-3.0Ag-0.5Cu solder ball to prove the impact of solder volume on interfacial reaction. The bigger the interfacial Cu_6_Sn_5_ grains and the thicker the Cu_6_Sn_5_ IMC layer were proven to be, the smaller the solder ball volume. With rising reflow times, the Cu_6_Sn_5_ grains average diameter, the thickness of the Cu_6_Sn_5_ IMC layer, and the consumed thickness of the Cu pad all improved.

## 6. Conclusions

In this review, we have summarized the causes that can impact the formation and growth of intermetallic compounds (IMCs), which are recognized to constitute an important role in joint structure. It can be concluded that:(1)The effect of minor alloying elements on the primary intermetallic and interfacial IMC can be substantial. The microstructure of eutectic alloy may be changed by the addition of a small amount of a given alloying element to the bulk microstructure. The alloying elements added to the solder also can influence the formation and growth of IMCs. This can decrease or increase the IMC’s growth/reaction rate and result in the formation of an extra reaction layer at the interface(2)The IMC layer is considerably affected by the surface finishes material during soldering. The thickness and composition of IMCs also greatly affected by surface finish layers that formed by a process called dissolution, where some amount of the surface metallization dissolves into the molten solder, and the formation of IMCs differs depending on the surface finish.(3)Different aging temperatures and time also influenced the thickness of interfacial intermetallic. Higher temperatures and longer aging times increase the IMC growth(4)The average thickness of the intermetallic for low solder volume is thicker than for high solder volume solders. This was due to the fact the Cu concentration in the solder grew quicker in smaller solder balls than in bigger ones.

## Figures and Tables

**Figure 1 materials-15-01451-f001:**
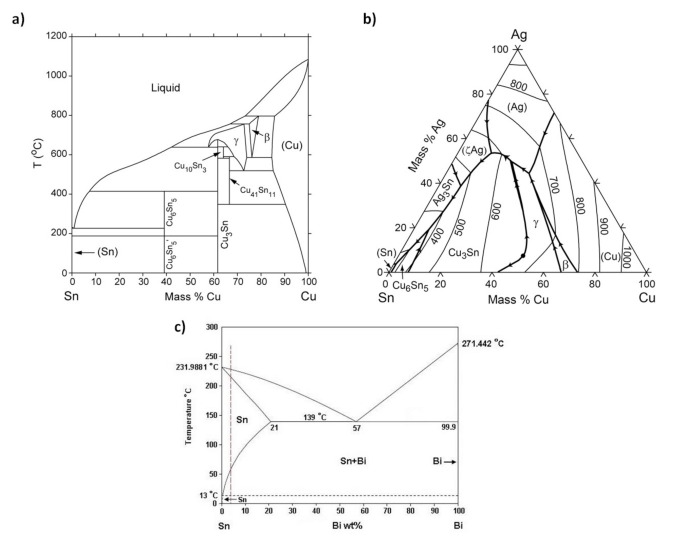
Phase diagram for (**a**) binary Sn-Cu, (**b**) ternary Sn-Ag-Cu and (**c**) binary Sn-Bi.

**Figure 2 materials-15-01451-f002:**
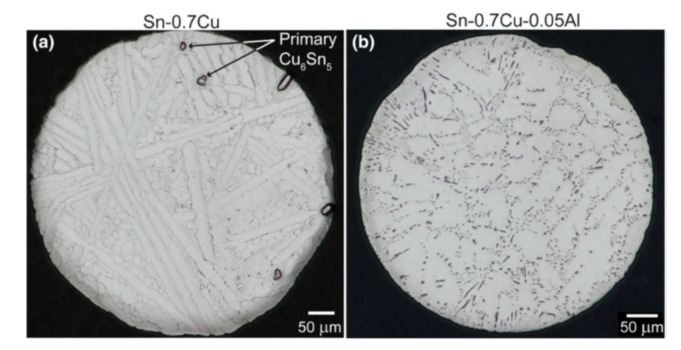
Microstructure images of (**a**) Sn-0.7Cu, (**b**) Sn-0.7Cu-0.05Al [14].

**Figure 3 materials-15-01451-f003:**
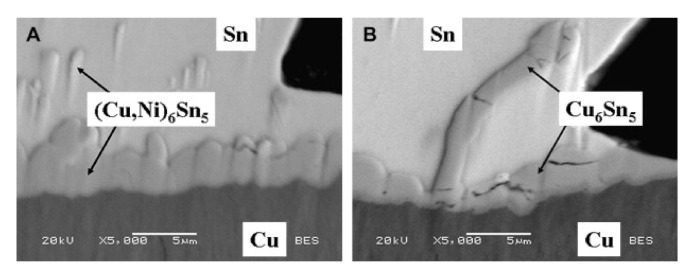
Typical cross-sectioned of solder joints; (**A**) Sn-0.7Cu-0.05Ni solder and (**B**) Sn-0.7Cu solder [24].

**Figure 4 materials-15-01451-f004:**
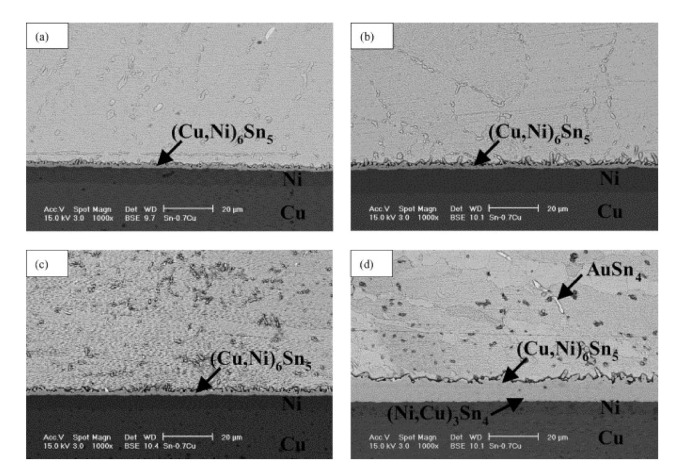
The Sn0.7Cu solder/Ni BGA SEM micrographs; Joints after aging for 100 days at a variety of temperatures: (**a**) 70 °C, (**b**) 100 °C, (**c**) 150 °C, and (**d**) 170 °C [85].

**Figure 5 materials-15-01451-f005:**
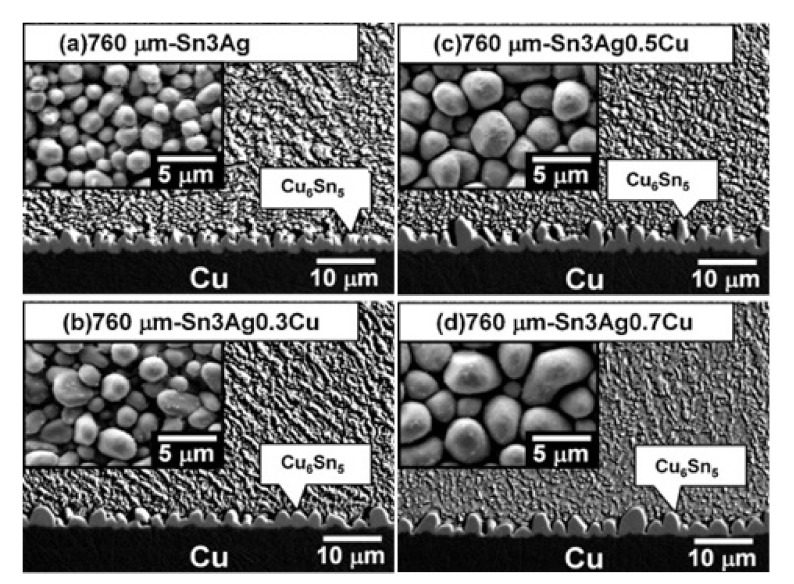
Cross-section and top-view (insets) micrographs showing the microstructure for the Sn3Ag0.5Cu solder with different ball sizes.(**a**) 960, (**b**) 760, (**c**) 500, and (**d**) 400 µm [106].

**Table 1 materials-15-01451-t001:** Influence of minor alloying elements to the IMC thickness in lead-free solder joints.

Solder Alloy	Element	Thickness of IMC	Ref
Sn-Ag-0.5Cu	Fe	Unchanged	[49]
Sn-3.0Ag-0.5Cu	Sb	Unchanged	[38]
Sn-3.0Ag-0.5Cu	Fe	Unchanged	[38]
Sn-3.0Ag-0.5Cu	In	Unchanged	[38]
Sn-3.0Ag-0.5Cu	Ge	Increase	[38]
Sn-3.9Ag-0.7Cu	La	Decrease	[47]
Sn-3.8Ag-0.7Cu	Nd	Decrease	[46]
Sn-2.8Ag-0.5Cu	Bi	Unchanged	[52]
	Mo	Decrease	[50]
Sn-3.8Ag-0.7Cu	Co	Decrease	[51]
	Zn	Decrease	[35]
Sn-0.5Ag-0.7Cu	Ga	Decrease	[45]
Sn-3.0Ag-0.5Cu	Co	Decrease	[44]
Sn-3.8Ag-0.7Cu	Zn	Decrease	[36]
Sn-4Ag	Zn	Decrease	[34]
Sn-0.7Cu	Al	Decrease	[43]
Sn-0.5Cu	Al	Decrease	[53]
Sn-0.7Cu	P	Increased	[42]
Sn-0.7Cu	In, Cr and Ni	Decrease	[39]
Sn-0.7Cu	Ni	Decrease	[26]
Sn-0.7Cu	Ni	Decrease	[30]
Sn-Cu	Ni	Decrease	[24]
Sn-0.7Cu-0.05Ni	Bi	Unchanged	[54]
Sn-0.7Cu-0.2Ni	In	Increased	[40]

**Table 2 materials-15-01451-t002:** The Sn-Ag-Cu IMC thickness on the different surface finish.

Surface Finish	Solder Alloy	IMC Thickness, µm	IMC Formation	Reflow Peak Temperature, °C	Dwell Time, Min	Ref
Cu-OSP	Sn-xAg-0.5Cu (x = 3.0, 4.0)	4–5	Cu_6_Sn_5_, Ag_3_Sn, Cu_3_Sn	230	20	[70]
	Sn-1.0Ag-0.5Cu	3–5	Cu_6_Sn_5_, Ag_3_Sn, Cu_3_Sn	250	-	[71]
	Sn-3.0Ag-0.5Cu	2.5–3	Cu_6_Sn_5_, Ag_3_Sn, Cu_3_Sn	300	-	[72]
	Sn-3.8Ag-0.7Cu	1.0–2.3	Cu_6_Sn_5_, Ag_3_Sn, Cu_3_Sn	270	2	[51]
	Sn-3.8Ag-0.7Cu	2.0	Cu_6_Sn_5_, Ag_3_Sn, Cu_3_Sn	244	1	[61]
ENIG	Sn-4.0Ag-0.5Cu	<2.0	Ni_3_Sn_4_, Ni_3_Sn_2_, Cu_6_Sn_5_, (Cu, Ni)_6_Sn_5_ Ag_3_Sn, (Ni, Cu)_3_Sn_4_, Ag_3_Sn	250	1	[62]
	Sn-3.5Ag-0.7Cu	<2.0	Ni_3_Sn_4_, Ni_3_Sn_2_, Cu_6_Sn_5_, (Cu, Ni)_6_Sn_5_, Ag_3_Sn, (Ni, Cu)_3_Sn_4_, Ag_3_Sn	255	1	[73]
	Sn-3.0Ag-0.5Cu	1.5–2	Ni_3_Sn_4_, Ni_3_Sn_2_, Cu_6_Sn_5_, (Cu, Ni)_6_Sn_5_, Ag_3_Sn, (Ni, Cu)_3_Sn_4_, Ag_3_Sn	250	-	[74]
	Sn-3.0Ag-0.5Cu	2–3	Ni_3_Sn_4_, Ni_3_Sn_2_, Cu_6_Sn_5_, (Cu, Ni)_6_Sn_5_, Ag_3_Sn, (Ni, Cu)_3_Sn_4_, Ag_3_Sn	245	1	[75]
ENEPIG	Sn-3.0Ag-0.5Cu	2.3	Ni_3_Sn_4_, Ni_3_Sn_2_, PdSn_4_, AuSn_4_, Cu_6_Sn_5_, (Cu, Ni)_6_Sn_5_, Ag_3_Sn, (Ni, Cu)_3_Sn_4_	250	1	[76]
	Sn-3.0Ag-0.5Cu	1.3–2.5	Ni_3_Sn_4_, Ni_3_Sn_2_, PdSn_4_, AuSn_4_, Cu_6_Sn_5_, (Cu, Ni)_6_Sn_5_, Ag_3_Sn, (Ni, Cu)_3_Sn_4_	260	1	[63]
	Sn-4.0Ag-0.5Cu	1.0–2.5	Ni_3_Sn_4_, Ni_3_Sn_2_, PdSn_4_, AuSn_4_, Cu_6_Sn_5_, (Cu, Ni)_6_Sn_5_, Ag_3_Sn, (Ni, Cu)_3_Sn_4_	230	-	[77]
Im-Ag	Sn-3.0Ag-0.5Cu	2–3	Cu_6_Sn_5_, Cu_3_Sn, Ag_3_Sn	250	1	[78]
	Sn-3.8Ag-0.7Cu-0.15Ni- 1.4Sb-3.0Bi, Sn-3.4Ag- 0.5Cu-3.3Bi, and Sn-3.8Ag- 0.8Cu-3.0Bi.	10–12	Cu_6_Sn_5_, Cu_3_Sn, Ag_3_Sn	250	1	[79]

**Table 3 materials-15-01451-t003:** Effect of alloying element and aging time on solder alloys interfacial IMCs.

Solder	Element	Aging Temperature, °C	Aging Time, h	Rate Constant of IMC Growth (µm/Day)	IMC Formation	Ref
Sn-3.0Ag-0.4Cu	In	100–180	1506	0.13	Cu_6_Sn_5_, Ag_3_Sn, Cu_3_Sn	[87]
Sn-3Ag-3Bi	In	120, 150 and 180	960	0.2	Cu_6_Sn_5_, Ag_3_Sn, Cu_3_Sn	[88]
Sn-2.8Ag-0.5Cu	Bi	150	336	0.5	Cu_6_Sn_5_, Ag_3_Sn, Cu_3_Sn	[52]
Sn-58Bi	Ce, La	80	168	0.5–0.79	Cu_6_Sn_5_, Cu_3_Sn	[92]
Sn-58Bi	Cr	100	240	0.19	Cu_6_Sn_5_, Cu_3_Sn	[93]
Sn-Bi	Ag	100	600	0.1	Cu_6_Sn_5_, Cu_3_Sn, Ag_3_Sn	[94]
Sn-2.5Ag-0.8Cu	Fe, Co and Ni	160	2000	0.44	Cu_6_Sn_5_, Ag_3_Sn, Cu_3_Sn	[91]
Sn-3.0Ag-0.5Cu	TiO_2_	190	720	0.37	Cu_6_Sn_5_, Ag_3_Sn, Cu_3_Sn	[95]
Sn-0.7Cu	Ni	170	2400	0.09	Cu_6_Sn_5_, Cu_3_Sn, (Ni, Cu)_6_Sn_5_	[85]
Sn-3.0Ag-0.5Cu	Bi and Er	150	500	0.14	Cu_6_Sn_5_, Ag_3_Sn, Cu_3_Sn	[3]
Sn-3.7Ag-0.7Cu	Bi and In	150	400	2.5	Cu_6_Sn_5_, Ag_3_Sn, Cu_3_Sn, (Ni, Cu)_3_Sn_4_,	[86]
Sn-3.0Ag-0.5Cu	Mo	180	480	0.7	Cu_6_Sn_5_, Ag_3_Sn, Cu_3_S	[96]
Sn-0.7Cu	Zn	150	480	0.35	Cu_6_Sn_5_, Ag_3_Sn, Cu_3_Sn, CuZn, Cu_5_Zn_8_	[90]
Sn-0.7Cu-0.06Zn	Ni	150	500	0.25	Cu_6_Sn_5_, Ag_3_Sn, Cu_3_Sn, (Ni, Cu)_3_Sn_4_, CuZn, (Ni, Cu)_6_Sn_5_	[97]
Sn99.3Cu0.7	Ge	150	720	0.22	Cu_6_Sn_5_, Cu_3_Sn	[98]
Sn-0.3Ag-0.7Cu	Mn	190	1152	0.10	Cu_6_Sn_5_, Ag_3_Sn, Cu_3_Sn	[99]
Sn-1.0Ag-0.5Cu	Fe and Bi	125	720	0.23	Cu_6_Sn_5_, Ag_3_Sn, Cu_3_Sn, FeSn_2_	[100]
Sn-4.0Ag-0.5Cu	Bi and Ni	175	2000	0.3	Cu_6_Sn_5_, Ag_3_Sn, Cu_3_Sn, (Ni, Cu)_3_Sn_4_, (Ni, Cu)_6_Sn_5_	[81]

## Data Availability

The data presented in this study are available on request from the corresponding author.

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
