# Peer review of "Formation and Growth of Intermetallic Compounds in Lead-Free Solder Joints: A Review"

_materials, 2022, doi:10.3390/ma15041451_

Round 1

Reviewer 1 Report

Although this review paper is well-written, I think this manuscript should be revised and addressed the points mentioned below in a satisfactory manner before publication in <Materials>.

- (P. 3 line 95) 0.7Cu Salleh, Mcdonald [9].

The authors should revise this sentence.

- (Fig. 2 caption) , should be changed to <and>.

- (P. 4 line 140,141,142) 5-10 m

The authors should recheck and revise this part.

- (P. 4 line 152) 1008 hours should be changed to 1008 h.

- The authors used <Amid> several times in this paper. Ex) Amid solidification

However, many researchers in this field used during instead of amid. Please revise the word.

- (P. 4 line 175) Delete the (Sn-Ni).

- (P. 4 line 175) The first phase of Cu-Sn interfacial intermetallic is formed nearest to the Cu interface and called “É›-phase” Cu3Sn intermetallic. Another phase of Cu-Sn, called “η-phase” Cu6Sn5, will form on top of the Cu3Sn intermetallic. The interfacial intermetallic formed between Ni and Sn at a much slower rate relative to the Cu-Sn intermetallic and is called the “δ-phase” Ni3Sn4 intermetallic. The Sn-Ni intermetallic is brittle than the Sn-Cu intermetallic.

The authors should cite the references.

- (P. 5 line 187) resulting in a coating that is 2-3 times thicker

Please explain it in more detail.

- (P. 8 line 311) Organic Solderability Pad (OSP) should be changed into Organic Solderability Preservative (OSP).

- (P. 8 line 320) (Cu, Ni)6Sn5 has been revealed to be more brittle than Cu6Sn5.

The authors should explain it in more detail by using previous papers.

- (P. 9 line 390 ~ 413) This paragraph is already mentioned. Please delete.

- (P. 11 line 460) Cu6Sn5is is

Please revise.

Author Response

Manuscript number : materials-1544135

Title: “Formation and growth of intermetallic compounds in lead-free solder joints: A review”

Dear Editor/Reviewer,                                                                                                    

Thank you for the editor and reviewers’ comments concerning our manuscript titled “Formation and growth of intermetallic compounds in lead-free solder joints: A review”. We have found these comments to be very helpful in revising and improving our manuscript.  After careful consideration and discussion with the co-authors, we have made some modifications and corrections to the submission, which we hope will meet your requirements for approval to be published in this journal. The revised manuscript has been submitted, and the response to the reviewers’ comments are as follows.

Reviewer(s)' Comments to Author:

Reviewer: 1

No

Comment

Correction

Page/Line

(As highlighted in manuscript)

1.

(P. 3 line 95) 0.7Cu Salleh, Mcdonald [9].

The authors should revise this sentence

Thank you for the suggestion. We have revised the sentence as below:

“The nucleation and development of the main Cu6Sn5 intermetallic were considerably affected when 0.05 wt% Ni was added to Sn-0.7Cu [10].

They also suggested that primary Cu6Sn5 developed at higher temperatures, which was smaller and more common in Sn-0.7Cu-0.05Ni/Cu joints compared to Sn-0.7Cu/Cu joints.”

Page 4, Line 100-104

2.

(P. 4 line 140,141,142) 5-10 m

The authors should recheck and revise this part.

Thank you for the suggestion. We have revised the sentence as below:

“Precisely, adding 0.25%RE elements resulted in one-third to half of the β-Sn grains transforming into smaller grains of 5-10 µm. When 0.5%RE elements were added, the initial β-Sn grains were refined to the point where virtually all of them were 5 - 10 µm in size”.

Page 5 Line 152-154

3.

(P. 4 line 152) 1008 hours should be changed to 1008 h.

Thank you for the comments. We have changed 1008 hours to 1008 h

Page 6, Line 205

4.

The authors used <Amid> several times in this paper. Ex) Amid solidification

However, many researchers in this field used during instead of amid. Please revise the word.

Thank you for the comments. We have revised the words “amid” to “during” in the manuscript.

All pages

5.

(P. 4 line 175) Delete the (Sn-Ni).

(P. 4 line 175) The first phase of Cu-Sn interfacial intermetallic is formed nearest to the Cu interface and called “É›-phase”Cu3Sn intermetallic. Another phase of Cu-Sn, called “η-phase”Cu6Sn5, will form on top of the Cu3Sn intermetallic. The interfacial intermetallic formed between Ni and Sn at a much slower rate relative to the Cu-Sn intermetallic and is called the“δ-phase” Ni3Sn4 intermetallic. The Sn-Ni intermetallic is brittle than the Sn-Cu intermetallic.

The authors should cite the references.

Thank you for the comments. We have deleted the (Sn-Ni) as suggested by the reviewer.

Thank you for the comments. We have revised the sentence as suggested by the reviewer as below:

“(Cu, Ni)6Sn5 has been revealed to be more stable than Cu6Sn5 at room temperature where the content of Ni is ~9 at. % [59,60]. We also added some references in the manuscript.

1.         Liu, S., et al., Inhibiting effects of the Ni barrier layer on the growth of porous Cu3Sn in 10-μm microbumps. Journal of Materials Science: Materials in Electronics, 2021. 32.

2.         Nogita, K. and T. Nishimura, Nickel-stabilized hexagonal (Cu,Ni)6Sn5 in Sn–Cu–Ni lead-free solder alloys. Scripta Materialia, 2008. 59: p. 191-194.

Page 9, Line 336-337

6.

(P. 5 line 187) resulting in a coating that is 2-3 times thicker

Please explain it in more detail.

Thank you for the comments. To not confuse readers, we have deleted that statement and revised the sentence as below:

“The Cu atom in Cu6Sn5 can be substituted with Ni atom, which changes the composition to (Cu, Ni)6Sn5”.

Page 6, Line 190-191

7.

(P. 8 line 311) Organic Solderability Pad (OSP) should be changed into Organic Solderability Preservative (OSP).

Thank you for the comments. We have changed Organic Solderability Pad (OSP) to Organic Solderability Preservative (OSP).

Page 9, Line  327

8.

(P. 8 line 320) (Cu, Ni)6Sn5 has been revealed to be more brittle than Cu6Sn5.

The authors should explain it in more detail by using previous papers.

Thank you for the comments. We have revised the sentence in the manuscript as below explaining on the (Cu, Ni)6Sn5 and added the references as below:

“(Cu, Ni)6Sn5 has been revealed to be more stable than Cu6Sn5 at room temperature where the content of Ni is ~9 at. % [59,60]”.

Page 9, Line 336-337

9.

(P. 9 line 390 ~ 413) This paragraph is already mentioned. Please delete.

Thank you for the comments. The paragraph has been deleted.

-

10.

(P. 11 line 460) Cu6Sn5is is

Please revise.

Thank you for the comments. The sentence has been revised as in the manuscript as below:

“The activation energy of Cu6Sn5 is lower than that of Cu3Sn”.

Page 13, Line 472

We thank you for the valuable suggestions made by the reviewers to improve our paper. All concerns and suggestions have been addressed in the above feedback. We would like to thank the editor in advance for considering our work.

Yours sincerely,

Assoc. Prof. Dr Mohd Arif Anuar Mohd Salleh

Nihon Superior Electronic Material Research Lab, Center of Excellence Geopolymer & Green Technology (CeGeoGTech), School of Materials Engineering, Universiti Malaysia Perlis (UniMAP), Taman Muhibbah, 02600, Arau, Perlis, Malaysia.

Reviewer 2 Report

Review of manuscript materials-1544135 submitted to Journal of Materials

“Formation and growth effect of intermetallic compounds in 2 lead-free solder joints: A review”

This work reviews the IMCs formation and growth factors such as alloying element, type of surface finish, solder volume, aging time and temperature in typical lead-free solder joints. In addition, this paper summarizes the factors that impacts the IMCs formation in the lead-free solder alloy. Authors gave introduction of the importance of effecting of IMCs on the lead-free solders and their possible applications in electronic devices. Through the investigation, the authors concluded that the effect of alloying elements and different types of surface finish on the primary intermetallic and interfacial IMC can be substantial. This can also decrease or increase the IMC's growth/reaction rate and result in the formation of an extra reaction layer at the interface. They also concluded that the characteristics of different aging temperatures and time influenced the thickness of interfacial intermetallic.     

I have the following comments regarding the manuscript:

The abstract does not reflect the content of the manuscript. So, it must be carefully re-written. Also, title must be more specified.

Many studies related to the field ignored and must be added to abroad the literature about this work???

"Nickel effects on the structural and some physical properties of the eutectic Sn-Ag lead-free solder alloy", Soldering & Surface Mount Technology, Vol. 31 No. 1, pp. 40-51.

Regarding the scientific and technological issues have been addressed in this work, further discussion should be added.

Overall English writing is good and the readability is good.

Author Response

Manuscript number : materials-1544135

Title: “Formation and growth of intermetallic compounds in lead-free solder joints: A review”

Dear Editor/Reviewer,                                                                                                    

Thank you for the editor and reviewers’ comments concerning our manuscript titled “Formation and growth of intermetallic compounds in lead-free solder joints: A review”. We have found these comments to be very helpful in revising and improving our manuscript.  After careful consideration and discussion with the co-authors, we have made some modifications and corrections to the submission, which we hope will meet your requirements for approval to be published in this journal. The revised manuscript has been submitted, and the response to the reviewers’ comments are as follows.

Reviewer(s)' Comments to Author:

Reviewer: 2

No

Comment

Correction

Page (As highlighted in manuscript)

1.

The abstract does not reflect the content of the manuscript. So, it must be carefully re-written. Also, title must be more specified.

Thank you for your comments. The abstract has been rewritten and the title has been changed as below:

Abstract

“Recently, research into the factors that influence the formation and growth of intermetallic compounds (IMCs) layer in lead-free solders has piqued interest, as IMCs play an important role in solder joints. The reliability of solder joints becomes more critical to the long-term performance of electronic products. One of the most important factors which are known to influence solder joint reliability is the intermetallic compound (IMC) layer formed between the solder and the substrate. Although the formation of an IMC layer signifies good bonding between the solder and substrate, its main disadvantage is due to its brittle characteristic. This paper reviews the formation and growth of IMCs in lead-free solder joints detailing the effect of alloying additions, surface finishes, aging time, aging temperature and solder volume. The formation and growth of these brittle IMCs were significantly effected by the factors and could be possibly controlled. This review may be used as a basis in understanding the major factors effecting the IMC formation and growth and relating it to the reliability of solder joints.”

Title:

“Formation and growth of intermetallic compounds in lead-free solder joints: A review”

Page 1, Line 24-35

Page 1, Line 2

2.

Many studies related to the field ignored and must be added to abroad the literature about this work???

"Nickel effects on the structural and some physical properties of the eutectic Sn-Ag lead-free solder alloy", Soldering & Surface Mount Technology, Vol. 31 No. 1, pp. 40-51.

Regarding the scientific and technological issues have been addressed in this work, further discussion should be added

Thank you for the comments. This review paper discussed several factors that could affect intermetallic compound formation. The effect of an alloying element, the effect of surface finish, the effect of aging time and temperature, and effect of solder volume have been a subtopic for this review. We have added more important references to the work as suggested by reviewers.

We have also added the discussion related to the paper suggested by the reviewer as below:

“The addition of Ni into Sn-Ag has been studied by Gumaan et. al [13]. They reported that the addition of Ni has been refined the particle size of β-Sn. Moreover, with 0.3 wt.% of Ni and distribution of Ag3Sn offer the potential benefit such as high strength, good plasticity and good mechanical performance.”

Page 4, Line 111-114

We thank you for the valuable suggestions made by the reviewers to improve our paper. All concerns and suggestions have been addressed in the above feedback. We would like to thank the editor in advance for considering our work.

Yours sincerely,

Assoc. Prof. Dr Mohd Arif Anuar Mohd Salleh

Nihon Superior Electronic Material Research Lab, Center of Excellence Geopolymer & Green Technology (CeGeoGTech), School of Materials Engineering, Universiti Malaysia Perlis (UniMAP), Taman Muhibbah, 02600, Arau, Perlis, Malaysia.

Reviewer 3 Report

Review of paper no. materials-1544135 titled Formation and growth effect of intermetallic compounds in lead-free solder joints: A review by Ramli et al.

This paper reviews the formation and growth of IMC layers in joints of lead-free solders with different substrates. Although some review articles on this topic are already available (see, e.g., http://dx.doi.org/10.1155/2013/123697) the present paper is sufficiently novel, as it discusses different solder compositions and substrates. The manuscript is publishable subject to minor revision.

1.There are several sentence structures with unclear meaning, e.g. “adding Al to Sn-0.7Cu size decrement (?) of Cu6Sn5” (line 107), “Sn-Ag-Co solders have IMC thicknesses about three times (higher/lower?) compared to binary Sn-Ag solders” (lines 218-219), etc. The paper should be proofread by a native speaker before resubmission.

2.The title should be shortened in the following manner: Formation and growth of intermetallic compounds in lead-free solder joints: A review

3.It is advisable to include most important phase diagrams in the paper, especially Sn-Cu and Sn-Ag-Cu, to explain the formation of specific IMCs.

4.Chapter 2 (alloying element effects) is very long, as it discusses both the IMC formation in the bulk solder and at the solder-substrate interface. It should be divided into two separate parts: one part should deal with the formation and growth of IMC in the bulk solder, and the other one with the solder-substrate interface.

5.Figures 1a and b should be annotated, as the difference between the microstructures is hardly visible.

6.Specify the chemical composition of Sn-Ag-Cu in the first row of Table 1.

7.The abbreviation “HASL” (line 336) is not explained in the manuscript.

8.The chemical composition of the IMC for each joint should be included in Tables 2 and 3.

9.It is advisable to include concrete rate constants for IMC growth instead of the thickness increase/decrease labels in Table 3.

10.Authors refer to Figures 8 and 9 (lines 489, 512); however, these figures are not present in the manuscript.

11.Concrete conclusions should be drawn from this work instead of general statements. They should be related to the alloying element effects, substrate effects and other factors discussed in the paper. Conclusions should be given point by point and numbered.

Author Response

Manuscript number : materials-1544135

Title: “Formation and growth of intermetallic compounds in lead-free solder joints: A review”

Dear Editor/Reviewer,                                                                                                    

Thank you for the editor and reviewers’ comments concerning our manuscript titled “Formation and growth of intermetallic compounds in lead-free solder joints: A review”. We have found these comments to be very helpful in revising and improving our manuscript.  After careful consideration and discussion with the co-authors, we have made some modifications and corrections to the submission, which we hope will meet your requirements for approval to be published in this journal. The revised manuscript has been submitted, and the response to the reviewers’ comments are as follows.

Reviewer(s)' Comments to Author:

Reviewer: 3

No

Comment

Correction

Page (As highlighted in manuscript)

1.

There are several sentence structures with unclear meaning,e.g. “adding Al to Sn-0.7Cu size decrement (?) of CuSn” (line107), “Sn-Ag-Co solders have IMC thicknesses about three times (higher/lower?) compared to binary Sn-Ag solders” (lines218-219), etc. The paper should be proofread by a native speaker before resubmission.

Thank you for your comments. We have rephrased the sentence in the manuscript as below. The paper also has been sent for proofread to a native speaker:

“They hypothesized that the addition of a small amount Al into Sn-0.7Cu solder alloy helps to reduce the Cu6Sn5 size while increasing the quantity numbers of Cu6Sn5 particles per unit volume”

“As a result, the thickness of IMC layer in Sn-Ag-Co solder will be increased about three times compared to binary Sn-Ag solders”

Page 4, Line 118-120

Page 7, Line 231-232

2.

The title should be shortened in the following manner: Formation and growth of intermetallic compounds in lead-free solder joints: A review

Thank you for your comments. The title has been changed as suggested by the reviewer as below:

“Formation and growth of intermetallic compounds in lead-free solder joints: A review”

Page 1, Line 1-2

3.

It is advisable to include most important phase diagrams in the paper, especially Sn-Cu and Sn-Ag-Cu, to explain the formation of specific IMCs.

Thank you for the comment. We have added the phase diagram in the manuscript as below:

Besides, the formation of intermetallics in solder alloy could be predicted by referring to the phase diagram as depicted in Figure 1.

Figure 1. Phase diagram for (a) binary Sn-Cu, (b) ternary Sn-Ag-Cu and (c) binary Sn-Bi.

Page 2, Line 80-83

4.

Chapter 2 (alloying element effects) is very long, as it discusses both the IMC formation in the bulk solder and at the solder-substrate interface. It should be divided into two separate parts: one part should deal with the formation and growth of IMC in the bulk solder, and the other one with the solder-substrate interface.

Thank you for your suggestion. We have divided chapter 2 into subchapters that explain the formation and growth of IMC in the bulk solder and another one with the solder-substrate interface as suggested by the reviewer.

The added subchapter in the manuscript as below:

2. 1 Formation and growth of IMC in the bulk solder

2. 2 Formation and growth of IMC at the solder-substrate interface

Page 2, line 72

Page 5, line 170

5.

Figures 1a and b should be annotated, as the difference between the microstructures is hardly visible.

Thank you for your comment. We decided to delete the figure and added the explanation with the references into the manuscript as below:

“Nogita [9] observed a considerable change in the nucleation pattern as well as behaviour of the Sn-Cu6Sn5 and Sn when Ni was added up to 1000 ppm to Sn-0.7Cu solder. In the absence of Ni, Sn-0.7Cu demonstrates that solidification progressed from the edge to the bulk solder alloy centre, a phenomenon is known as the 'wall mechanism”.

Page 3, line 97-104

6.

Specify the chemical composition of Sn-Ag-Cu in the first row of Table 1.

Thank you for your comment. We have added the composition of Sn-Ag-Cu in the first row of table 1.

Page 9, line 319

7.

The abbreviation “HASL” (line 336) is not explained in the manuscript.

Thank you for the comments. The explanation on the abbreviation of “HASL” has been added in the manuscript as below:

“discovered that immersion tin coating produced a smooth and homogeneous IMC layer when contrasted to copper plated with hot air solder levelling (HASL), OSP, or immersion silver”

Page 9, line 351-353

8.

The chemical composition of the IMC for each joint should be included in Tables 2 and 3.

Thank you for your comment. We had added the composition of IMC for each joint in Table 2 and Table 3 as suggested by the reviewer.

Page 11, line 425

Page 15, line 542

9.

It is advisable to include concrete rate constants for IMC  growth instead of the thickness increase/decrease labels in Table 3.

Thank you for your comment. We had added the rate constant for each solder in Table 3 as suggested by the reviewer.

Page 15, line 542

10.

Authors refer to Figures 8 and 9 (lines 489, 512); however, these figures are not present in the manuscript.

Thank you for your comment. We have deleted Figures 8 and Figures 9 in the manuscript.

-

11.

Concrete conclusions should be drawn from this work instead of general statements. They should be related to the alloying element effects, substrate effects and other factors discussed in the paper. Conclusions should be given point by point and numbered.

Thank you for the comments. The conclusion has been rewritten as suggested by the reviewer  in the manuscript as below:

In this review, we have summarized the causes that can impact the formation and growth of intermetallic compounds (IMCs), which are recognized to constitute an important role in joint structure. It can be concluded that:

1)     The effect of minor alloying elements on the primary intermetallic and interfacial IMC can be substantial. The microstructure of eutectic alloy may be changed with the addition of a small amount of a given alloying element to the bulk microstructure. The alloying elements added to the solder also can influence the formation and growth of IMCs. This can decrease or increase the IMC's growth/reaction rate and result in the formation of an extra reaction layer at the interface.

2)     The IMC layer is considerably affected by the surface finishes material during soldering. The thickness and composition of IMCs also greatly affected by surface finish layers that formed by a process called dissolution, where some amount of the surface metallization dissolves into the molten solder, and the formation of IMCs differs depending on the surface finish.

3)     Different aging temperatures and time also influenced the thickness of interfacial intermetallic. Higher temperature and longer aging time increases the IMC growth.

4)     The average thickness of the intermetallic for low solder volume is thicker than for high solder volume solders. This was due to Cu concentration in the solder grew quicker in smaller solder balls than in bigger ones.

Page 16, line 597-617

We thank you for the valuable suggestions made by the reviewers to improve our paper. All concerns and suggestions have been addressed in the above feedback. We would like to thank the editor in advance for considering our work.

Yours sincerely,

Assoc. Prof. Dr Mohd Arif Anuar Mohd Salleh

Nihon Superior Electronic Material Research Lab, Center of Excellence Geopolymer & Green Technology (CeGeoGTech), School of Materials Engineering, Universiti Malaysia Perlis (UniMAP), Taman Muhibbah, 02600, Arau, Perlis, Malaysia.

Round 2

Reviewer 1 Report

As the paper has been improved I recommend that it will be published in the <Materials>.

Reviewer 2 Report

After the corrections it should be accepted